# A Proposal of Bioinspired Soft Active Hand Prosthesis

**DOI:** 10.3390/biomimetics8010029

**Published:** 2023-01-11

**Authors:** Alejandro Toro-Ossaba, Juan C. Tejada, Santiago Rúa, Alexandro López-González

**Affiliations:** 1Computational Intelligence and Automation Research Group (GIICA), Universidad EIA, Envigado 055428, Colombia; 2Department of Engineering Studies for Innovation, Universidad Iberoamericana, Ciudad de México 01219, Mexico; 3School of Basic Sciences, Technologies and Engineering, Universidad Nacional Abierta y a Distancia, Medellín 050012, Colombia

**Keywords:** soft robotics, hand prosthesis, biomimetic, myoelectric control, neural networks

## Abstract

Soft robotics have broken the rigid wall of interaction between humans and robots due to their own definition and manufacturing principles, allowing robotic systems to adapt to humans and enhance or restore their capabilities. In this research we propose a dexterous bioinspired soft active hand prosthesis based in the skeletal architecture of the human hand. The design includes the imitation of the musculoskeletal components and morphology of the human hand, allowing the prosthesis to emulate the biomechanical properties of the hand, which results in better grips and a natural design. CAD models for each of the bones were developed and 3D printing was used to manufacture the skeletal structure of the prosthesis, also soft materials were used for the musculoskeletal components. A myoelectric control system was developed using a recurrent neural network (RNN) to classify the hand gestures using electromyography signals; the RNN model achieved an accuracy of 87% during real time testing. Objects with different size, texture and shape were tested to validate the grasping performance of the prosthesis, showing good adaptability, soft grasping and mechanical compliance to object of the daily life.

## 1. Introduction

The loss of upper extremities, specifically transradial loss, due to various factors such as occupational accidents, diseases, congenital problems, war, and so on, it is a common event [1,2]. Although there are a large number of prostheses to solve this problem, most of them fall short in their motor skills due to their simplicity, or on the contrary, their complexity makes them so expensive that it is impossible for a person to acquire them [3,4].

Soft robots are biologically inspired machines [5] using softness and compliance to interact with the environment were body deformations are used for optimal object manipulation and locomotion [6]. Bioinspired soft robotics allows robotic systems to have a more natural and adaptable behavior that permit adaptive interactions with unpredictable environments. There are three key points that allow a soft robot to obtain these advantages: design, material and actuation [7,8].

In the past decade, multiple authors have studied different actuation mechanisms that allow to exploit the advantages of soft robots; these actuators can change its shape by converting physical or chemical energy into mechanical deformation, that can be use as actuation mechanism for a soft robot. The most common actuation mechanism is via the pressurization of a fluid that causes the deformation of the actuator; among these actuators, the most widely used are pneumatic actuators with internal chambers that allow to control the direction of the deformation, also known as PneuNets [9,10,11,12,13]; other researchers have developed fluidic actuators based on an electro-conjugate fluid (ECF), which allows to create pressurization of the actuator using a ECF jet (hydraulic pressure source) that generate the deformation of the actuator [14,15]. Other notable actuators are electroactive polymers (EAPs) which generate movement when an electric field is applied to them [16,17]. Lastly, multiple authors have also proposed cable driven soft actuators, which allow to control the motion of the actuator by retracting cables embedded in its structure [18,19,20]

The advantages present in soft robots and soft actuators make them promising in the fields of prosthesis and orthotic devices, this is because soft robotics systems allow a more natural look and movement, giving greater compliance and adaptability to the user. Multiple researches have proposed different soft robotic hands, some of the most common soft robotic hands operate using cable driven fingers that allow to perform the different movements and grasping patterns [21,22,23,24,25]. Another common actuation method in soft robotic hands is the use of pneumatic fingers, this fingers are manufactured to be hollow on the inside, so they can perform bending motions when pressurized [26,27,28,29,30]. On one hand, pneumatic actuated prosthesis are often loud, expensive, and less portable due to the necessary pneumatic system for the actuation, hindering their implementation outside of a laboratory setting, on the other hand, most of the cable driven soft prosthesis lack the anatomical properties that made human hands unique, limiting their dexterity.

This research presents a dexterous bioinspired active hand prosthesis based on soft robotics operated by electromyography (EMG) signals. The main contribution of this research is the use of the human hand anatomy for the design and manufacturing of the soft prosthesis; this design, allows the system to have the biomechanical properties of the human hand, while having a low mechanical complexity of the overall system, thus reducing costs and manufacturing complexity, but allowing the hand to perform with with compliance and move like a human hand, adapting to objects of different sizes, shapes and textures. This concept allows an easy scalability of the prosthesis for future research and brings this type of system closer to having a more natural movement and appearance. The organization of the paper is as follows: Section 2 presents the design of the hand prosthesis. Section 4 shows the materials selection and fabrication process. Section 5 describes the control implementation for the prosthesis. Section 6 shows test results of the bioinspired soft active hand prosthesis. Finally, some conclusions and future work are presented in Section 7.

## 2. Design

The hand is extremely complex and intricate part of the human body that allow us to support, manipulate and grasp different objects [31,32]. This outstanding capacity of the human hand is given by its musculoskeletal configuration, which allow it to be remarkably mobile and adaptable as it conforms to the shapes of the objects grasped by it [33,34].

The skeletal architecture of the fingers is composed of four bones, one bone called the metacarpal and three phalanges, and the thumb is only composed of one metacarpal and two phalanges. Each one of the metacarpals articulates with the wrist bones (Carpals) forming the Carpometacarpal joints (CMC), which have as primary movement the palmar abduction, this joint is particularly mobile in the thumb, allowing its rotation and also abduction. The next joint in each finger is the metacarpophalangeal (CMP) which connects the metacarpals with the proximal phalanx. The following two joints are the ones that link the phalanx and are the proximal interphalangeal joint (PIP) and the distal interphalangeal joint (DIP); this three joints are the ones that allow the flexion and extension movements of each one of the fingers.

This skeletal configuration is commonly represented as a 23-DOF model that includes all the possible movements of the hand [28,35,36,37,38]; even though this number of DOF its what gives the hand its complexity, it’s difficult with the current technology to copy its biological performance without sacrificing portability, mechanism simplicity, easy of control and cost. The proposed bioinspired soft robotic hand aims to preserve the mechanical advantages of the skeletal architecture of the hand, while simplifying its kinematic complexity, making it a cost-effective and efficient prototype. Figure 1 shows the hand structure and the simplified 14-DOF model of the proposed soft robotic hand.

The bones of the hand are arranged in a way that they form three arches, two transverse and one longitudinal [31,33,34], these three arches, particularly the longitudinal arch, allow the palm to cup or flatten to accommodate objects of different shapes and sizes [33,39] making it more adaptable and complaint when performing grasping tasks. The proposed biomimetic structure for the prosthesis was able to imitate these three skeletal arches providing the system with the aforementioned biomechanical advantages that only the human hand has. This configuration can be seen in Figure 2.

The human hand also has structural components called ligaments and tendons sheats. These structural components generate balanced forces in the hand that create biomechanical stability across the skeletal structure and allow the correct movement of the different fingers of the hand [33]. Both of these components encloses, compartmentalizes and restrains the joints and tendons providing stability to the skeletal structure and constraining the movement of the fingers so they do not collapse laterally [31,33].

The bioinspired soft prosthesis proposed in this research emulates this musculoskeletal components in order to stabilize the skeletal structure (Bones) of the prosthesis and exploit the biomechanical advantages present in the human hand. The proposed bioinspired prosthesis emulates the digital collateral ligaments, which restrain the lateral movement of the MCP, PIP and DIP joints allowing them to move only in the direction of flexion and extension, these ligaments are the main stabilizer components, allowing the skeletal structure to maintain its integrity during different movements. The bioinspired prosthesis also emulates the tendon sheats, also known as finger pulleys, these sheats restrained the tendons to keep them close to the bone so they maintain a constant moment arm, preventing them from “bowstringing” across the joints [33]; these tendon sheats were formed by two elements, first a rubber band located in the proximal phalanx of each one of the fingers and second, a layer of Smooth-On Ecoflex 00-10 silicone rubber that covered each finger and worked as a constraining element for the tendon, and as a soft covering for the hand. In the case of the wrist bones and CMC joints, they were attached together with a “rigid ligament” that allowed very limited mobility in the CMC joints, the purpose of the “rigid ligament” was to support and stabilize the skeletal structure in the wrist and CMC joints, while allowing a small amount of palmar abduction.

Another fundamental musculoskeletal component is the tendon, the tendons are elements attached to some bones in each one of the fingers that pull them generating their movement [33]. In the proposed bioinspired prosthesis, the flexor digitorum profundus tendon, in charge of the flexion of the MCP, PIP and DIP joints, and the flexor pollicis longus, in charge of the flexion of the MCP and IP joints of the thumb, were emulated using a Nylon line, this line was attached to the distal phalanges of each finger; this mechanical set up allowed to actuate all three joints (two joints in the case of the thumb) simultaneously using an underactuation approach for the joints, this approach reduces the kinematic complexity of the prosthesis (reduces number of DOF) but preserves the biomechanical advantages of the human hand, because it allows the complete flexion and extension of the fingers. The tendons implemented in the bioinspired prosthesis were guided and restrained by a cavity in the metacarpal bones and the tendon sheats present in phalanges in order to guide the movement of each one of the fingers correctly. Figure 3 shows the aforementioned musculoskeletal elements implemented in the proposed soft robotic prosthesis.

Each one of the Nylon tendons seen in Figure 3 was actuated using a Micro Gearmotor N20 of 20 RPM and 1.5 kg. The motors were placed in the prosthesis socket and actuated each one of the tendons using a 3D printed pulley. This configuration allows the prosthesis to be modular, improving the scalability of the system.

## 3. Socket Design

For a hand prosthetic system to be feasible and implementable, a key characteristic is that it needs to be self-contained, this means that its electronic components, battery, and actuation motors should be housed inside the structure of the prosthesis. In the case of a transradial prosthesis, these components can be housed inside a socket, which is the element that attaches to the stump and is also responsible of supporting the soft prosthetic hand. There are two main characteristic that validate the feasibility of the socket. First, its dimensions must be close to the natural limb dimensions; and second, its weight must also be close to the weight of the forearm and hand.

In this work a socket is proposed so the soft prosthetic hand is feasible and implementable in a real prosthetic system. With this in mind, the proposed socket was designed taking into account the average forearm and hand dimensions; the average lenght of the forearm from the elbow joint to the wrist for a male is 290 mm and the average hand lenght is 194 mm [40]. The proposed socket was designed with a lenght of 220 mm considering that the subject may have part of the forearm, the socket also has an adjustable suspension sleeve with and standard lenght of 46 mm; this socket is attached to the proposed soft robotic hand which has a lenght of 180 mm; resulting in a total lenght for the prosthetic system of 446 mm, which is within the average lenght of the forearm and hand. The dimensions of the proposed socket and prosthetic hand can be seen in Figure 4.

The proposed socket is able to contain the five N20 Micro Gearmotors used in the actuation of each finger, a 7.4 V— 1000 mAh LiPo battery, the motors drivers and a Raspberry Pi Zero. The distribution of the components inside the socket can be seen in Figure 5a,b shows the closed socket. One important thing to note, is that the space within the socket can be further optimized using custom electronic boards with SMD components.

As mentioned before, one key element is the weight of the hand prosthetic system. The average weight of the forearm and hand is 2.52% of the body weight [41,42]. Assuming a male with a body weight of 70 kg, the resulting a forearm and hand weight is 1.76 kg. The proposed soft prosthetic hand system (hand plus socket) has an approximate weight of 0.5 kg including motors, battery, electronic components and considering it is manufactured in ABS material; thus, meeting the weight requirements necessary for the implementation and feasibility of this prosthetic system system.

## 4. Manufacturing

To fabricate the bioinspired soft robotic prosthesis each one of the hand bones were modeled using CAD software based on a human hand model. Due to the complexity of attaching the ligaments on the bone surface, each bone was designed with cavities that allowed to tie the rubber ligaments to the bones, this approach is useful to generate better unions at the joints and a more stable structure; the metacarpal bones also included an extra cavity that was used as guide for the nylon tendons. Once the hand bones were designed, they were 3D printed in PLA (Polylactic Acid) polymer material using fused deposition modeling (FDM), an infill density of 20% was used to print the bones, this value allowed the bones to be light but also to have a good mechanical resistance for the proposed application. Figure 6 shows the CAD model of one of the bones and the 3D printing process.

Once the bones were printed, they were joined together with rubber bands emulating the collateral ligaments of the three main finger joints (MCP, PIP and DIP) as mentioned in the design, the ligaments were passed through the cavities in the bones, then they were tied together to join the two bones that form each one of joints, as mentioned in Section 2, this setup stabilizes the joints and the skeleton structure, but it also allows the joints to be flexible, since the union between elements of the joint is not rigid and does not have a fixed rotation point. The wrist bones were also joined using rubber bands which passed through their interior and attached to each one of the metacarpals, this was done to increase the overall stability of the skeleton structure and to easy the assembling process.

Once the hand skeleton joined together, additional rubber bands were placed around the proximal phalanges to form the tendon sheats as mentioned in Section 2. Lastly 0.8 mm Nylon tendons were attached to the distal phalanges of each finger and were guided through the tendon sheats and metacarpals cavities; this configuration allowed to move all three joints simultaneously and constrain the fingers to only perform flexion and extension movements. Once the tendons were guided and attached in each finger, the were attached to the N20 Micro Gearmotors using 3D printed pulleys, each one of the pulleys had a radius of 1 cm, the pulleys were made with PLA polymer material and an infill density of 20%.

The wrist bones were covered with epoxy putty to give the additional support considering that the prosthesis don’t have an wrist movement, the epoxy also support the metacarpals at the CMC joint, but allows some movement so the prosthesis can perform palmar abduction. The fingers, wrist and joints were then covered with four layers of Smooth-On Ecoflex 00-10 silicone rubber, this cover was made in order to give additional support to the joints and tendons, working as “ligament material” and also as “tendon sheath”, this cover also gives the hand a soft texture when grasping objects.

## 5. Control

In order to control the soft robotic biomimetic prosthesis, a myoelectric control system was proposed due to its potential to provide an intuitive and natural control over the system by the user [43,44]. The myoelectric control system allows the user to move the prosthesis by capturing electromyography (EMG) signals from its own arm and processing into real movements for the robotic hand. EMG signals indicate the electric activity in the muscles due to the contraction of the muscle fibers, an is an effective method to map the muscle activity during dynamic movements and static positions [45]. The myoelectric control system is divided in three different layers: acquisition, control, and power (see Figure 7).

In the acquisition stage, the electromyography signals are acquired by four MyoWare Muscle Sensors. Each sensor rectifies and integrates the EMG signal and outputs the EMG signal envelope. The envelope is acquired by a Teensy 3.5 development board with a sampling frequency of 1 kHz; then it sends the acquired EMG data to the control stage (PC) via serial communication. In the PC, a neural network predicts which gesture is being performed and then passes the information to a Raspberry Pi Zero via WiFi. Finally, the Raspberry Pi Zero outputs the control signals for each motor driver, depending on the gesture that is being predicted. The motor drivers control the N20 Micro Gearmotors that moves each one of the fingers in the prosthesis. Each finger can also be controlled manually using the PC keyboard in case different gestures or movements must be performed. The output of the neural network is given by $G={\left[g_{1},\ldots ,g_{i}\right]}^{T}$, where $g_{i}\in \left[0,1\right]$, and i={1,2,…,n} is the number of gestures. Then, each motor is moved accordingly with the following equation
(1)$$M=KM_{A}G,$$
where $M_{A}\in R^{5\times n}$ is the allocation matrix and $K=\mathrm{diag}(k_{i})$ with $k_{i}\in R$ is a calibration matrix used to adjust the force and time to close and open each finger.

The four MyoWare EMG sensors were placed on top of the forearm targeting the muscles involved in the performance of the different gestures, particularly the flexor and extensor muscles of the fingers. Figure 8 depicts the location in which each one of the electrode pairs were placed.

### Recurrent Neural Network (RNN) Classifier

An LSTM-RNN architecture was proposed to perform the gesture classification task [46]. RNN was selected because this type of deep learning models are ideal for pattern recognition in time series signals due to its capacity to retain information of time sequences [46,47,48,49,50].

The LSTM-RNN architecture is formed by an input layer of 200×4 (samples × channels), a fully connected layer with 32 units and tanh activation function, a layer of Long Short-term Memory (LSTM) with 16 units, followed by a fully connected layer and an output layer with softmax activation to perform the multi-class classification (see Figure 9). Different investigations have found that this type of hybrid architecture outperforms architectures that only implement recurrent layers [51,52].

In order to train the neural network, EMG signals from five different gestures were recorded using the four MyoWare sensors, the gesture were open hand, close hand, lateral pinch, signaling sign and rock sign, these gestures can be seen in Figure 10. The EMG data was collected in three recording sessions; in which a total of 60,000 samples were collected for each gesture (20,000 samples per recording), creating with all gestures 300,000 samples of EMG data. This data was split in windows of 200 samples that served as input for the neural network, forming a dataset of 1500 windows. This dataset was randomly shuffled and split, forming a training/validation dataset of 1000 windows and a testing dataset of 500 windows.

A 10-Fold Cross Validation was performed to train and validate the classifier using the training/validation dataset. In this process, the LSTM-RNN model obtained an average accuracy of 0.9379±0.0147 in training and 0.9070±0.0347 in validation. The model was trained with the 1000 windows and then tested on the 500 windows dataset, in which it obtained an accuracy of 0.8780.

## 6. Results

### 6.1. EMG Control

The EMG control system was tested in real-time after training it. A total of four testing sessions were performed, in each session each gesture was maintained for 20 s. The average accuracy across these real-time sessions was 0.8729±0.0694.

The system turned out to be sensitive to variations in the position of the EMG sensors in different acquisition sessions, this sensitivity is reflected in the accuracy variance in the real-time testing. This sensitivity in EMG pattern classifiers is still one of the main challenges in EMG control systems for prosthetic devices [44,53,54], specially in systems with a low number of EMG channels. Some approaches to correct this issue could be, increase the number of EMG acquisition channels to have a better mapping of the muscle activity when the gestures are performed and increase the training data performing more recording sessions in order to allow the classifier to generalize better to these small variations in the EMG sensors.

### 6.2. Object Grasping

The human hand has the capacity to perform many complex and intricate movements that allow humans to engage in repetitive tasks and manipulate different objects. The human hand can perform a variety of prehensile and nonprehensile movements like grasping object, hook or spread [55].

The two fundamental patterns of prehensile hand function are: the power and the precision grip [33,56]. The power grip is performed with the fingers flexed at all three joints (MCP, PIP and DIP) so that the object is held between the fingers and the palm. The precision grip, also termed precision handling [57], consists in the manipulation of small object between the thumb and the flexor aspect of the fingers. Unlike the power grip, the precision handling does not involve a forceful griping of the object, rather its manipulation in a finely controlled manner [33].

The dexterity and grasping ability of the bioinspired prosthesis was tested performing the two fundamental grips mentioned before. The testing consisted in grasping different daily life objects that had different textures, shaped and stiffness, the objects were a jar, a rubber ball, a sauce bottle, a plastic cup, a screwdriver, a jar handle, a key chain, a pencil and a plastic card. These grasping tasks allowed to determine the adaptability and compliance of the prosthesis.

Figure 11 shows soft robotic hand performing a power grip with objects with different shapes and sizes. The prosthetic hand is capable of adapting to various shapes like cylindrical and spherical shapes, it has the capacity to hold firmly cylindrical shaped objects and also to softly hold easily deformable objects like a plastic cup, a rubber ball and a sauce bottle; showing good mechanical adaptability to different materials, geometries and shapes.

Figure 12 shows precision handling of different objects of the daily life, particularly shows the lateral pinch; this is an useful precision handling pattern that allow the hand to grab small object like keys and cards. The hand is also capable of performing the pulp pinch and some forms of “dynamic tripod”, for example, grabbing a pen.

The precision handling ability of the hand can be further enhanced by implementing thumb abduction and adduction, this movements will allow the soft robot arm to perform another precision handling patterns like the tip pinch and palmar pinch.

The soft robotic arm showed good adaptability and mechanical compliance to the the different objects without the need of any position sensing feedback.

### 6.3. Force and Speed

The grasping force of the proposed soft robotic hand was tested using the Hoggan Scientific microFET Digital HandGRIP dynamometer as it can be seen in Figure 13. In this test the prosthesis achieved a maximum grasping force of 12 N, which was enough to hold multiple objects of the daily life as seen in previous sections. It’s important to note that although the maximum force is limited by the N20 Micro Gearmotors, the structural capacity of the prosthesis is higher, the Nylon tendons are capable of withstanding up to 310 N.

The speed of actuation of the prosthesis was also tested. The EMG acquisition stage takes 200 ms in collecting an EMG window to be processed by the RNN model, once the window is acquired, the computation of the RNN model take approximately 50 ms; finally, the actuators take approximately between 2.8 s and 3 s to perform the desired gesture. So the system has a total time of actuation of approximately 3 s to 3.3 s. Like the force, the actuation time is constrained by the N20 Micro Gearmotors RPMs.

## 7. Conclusions and Future Work

In this work, a bioinspired 14-DOF soft robotic hand is presented. The proposed hand emulated the skeletal structure of the human hand and the main musculoskeletal components in order to take advantage of the biomechanical properties of the biological design of the human hand. This bioinspired design proved to be adaptable and compliant with objects of different geometries and sizes, allowing the soft robotic hand to perform the fundamental prehensile functions.

An EMG control system based on an RNN architecture for gesture recognition was also proposed. This EMG system allowed to perform five different gestures on the prosthesis in an intuitive manner, it obtained an accuracy of 0.8729±0.0694 in real-time operation using only four EMG channels to map the muscular activity of the forearm.

Future work includes adding thumb abduction to the soft robotic hand so it can perform more precision handling movements like the tip pinch and palmar pinch; reinforcing the lateral ligaments in the joints to provided greater stability; implementing Micro Gearmotors with greater RPMs and torque; increasing the number of EMG channels to better map the muscle activity in the forearm in order to decrease the LSTM-RNN model sensitivity, increase its accuracy, increase the number of gestures and further optimization of the LSTM-RNN model.

## Figures and Tables

**Figure 1 biomimetics-08-00029-f001:**
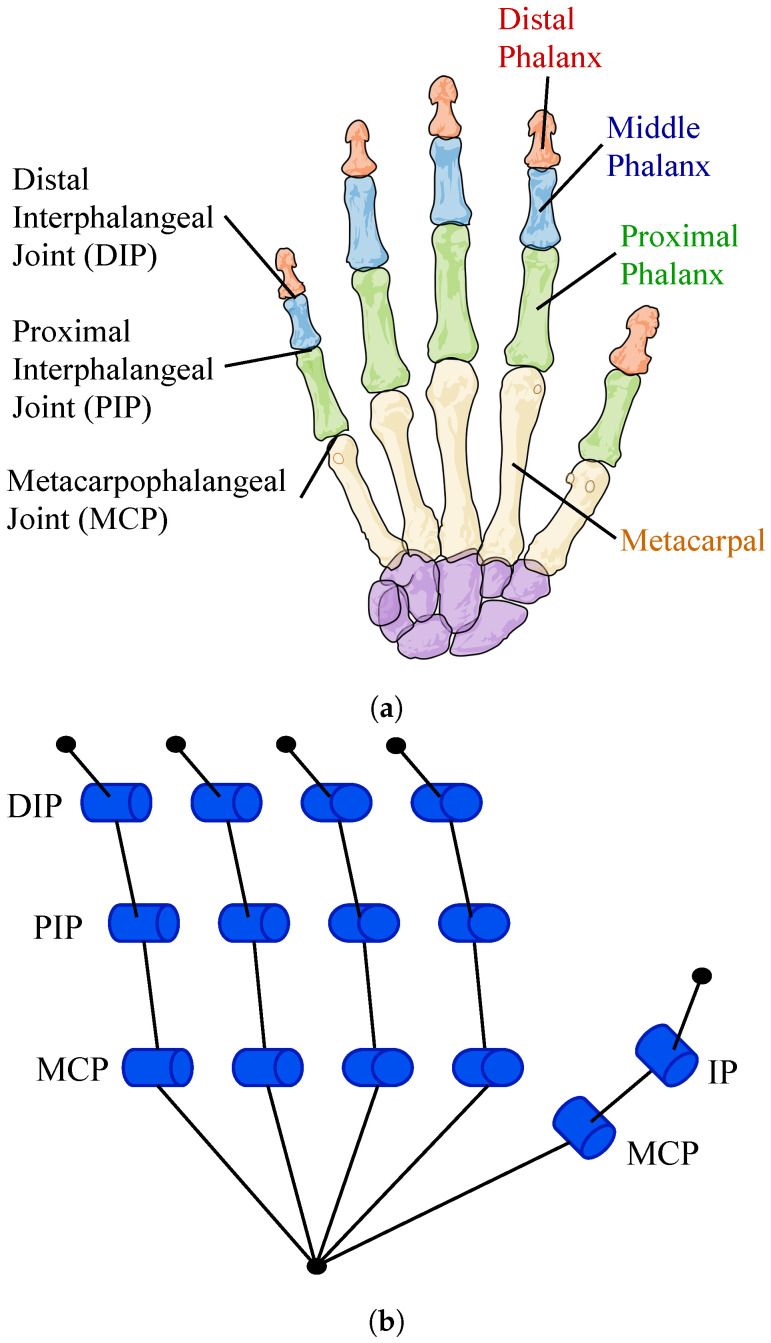
Human hand model. (**a**) Bones and joints of the human hand. (**b**) Simplified 14-DOF model of the proposed soft robotic prosthesis.

**Figure 2 biomimetics-08-00029-f002:**
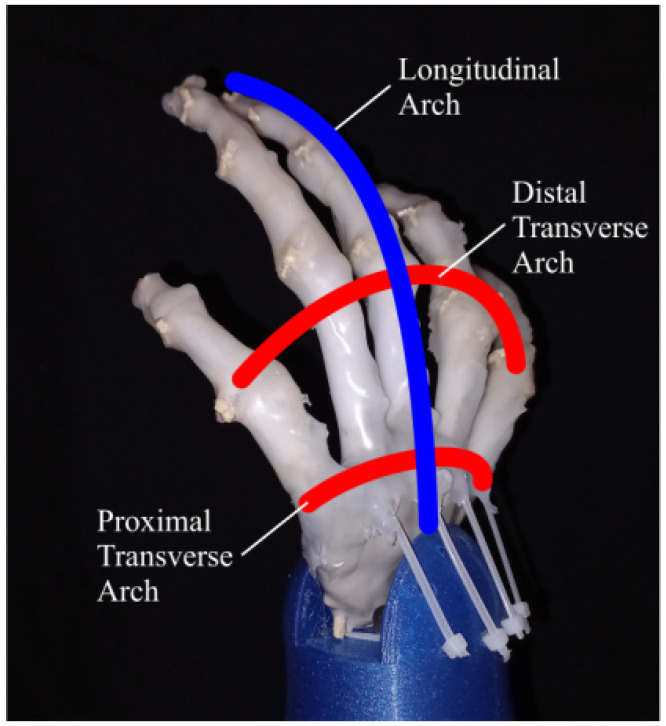
Skeletal arches of the hand.

**Figure 3 biomimetics-08-00029-f003:**
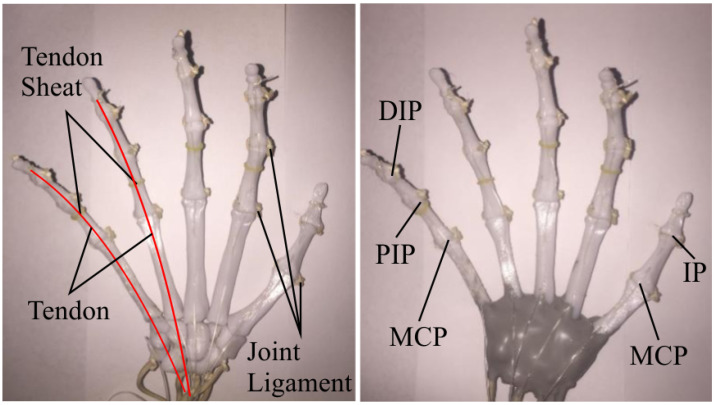
Musculoskeletal components and biomimetic structure of the soft robotic prosthesis.

**Figure 4 biomimetics-08-00029-f004:**
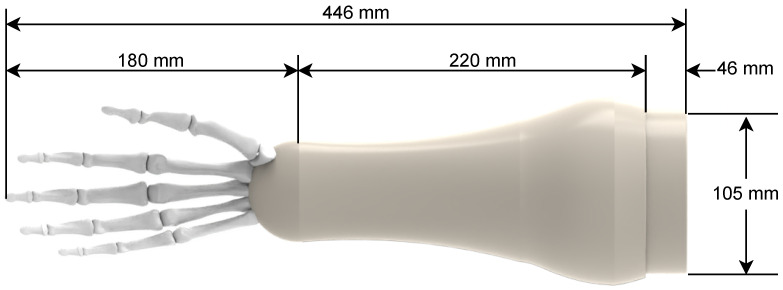
Socket dimensions.

**Figure 5 biomimetics-08-00029-f005:**
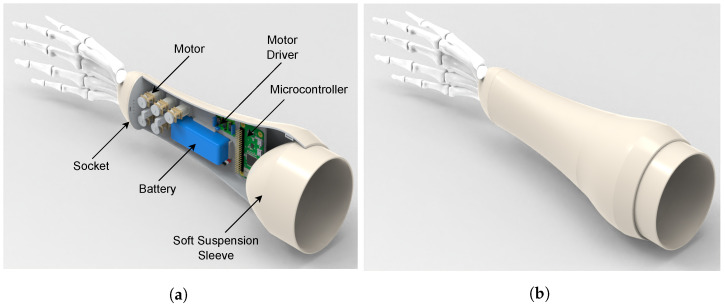
Soft prosthesis socket design. (**a**) Socket components. (**b**) Closed socket.

**Figure 6 biomimetics-08-00029-f006:**
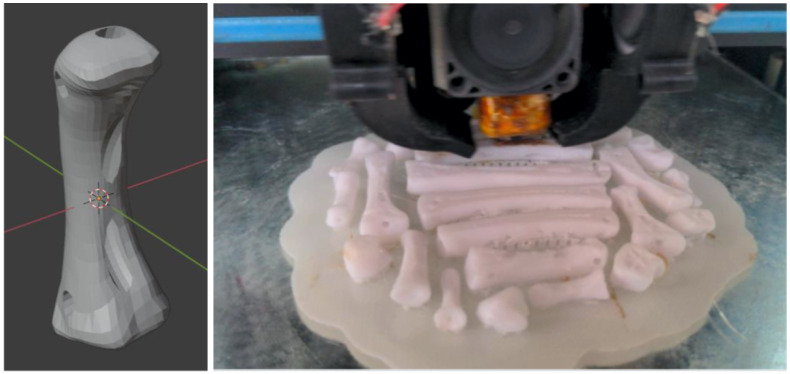
Metacarpal bone CAD model and 3D printing process of hand bones.

**Figure 7 biomimetics-08-00029-f007:**
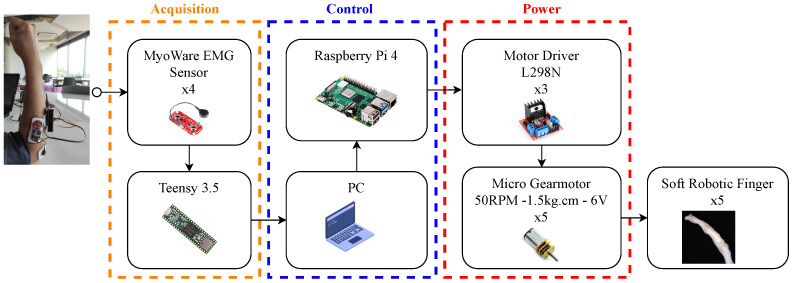
Hardware block diagram of the soft hand prosthesis. Acquisition, control and power layers of the system.

**Figure 8 biomimetics-08-00029-f008:**
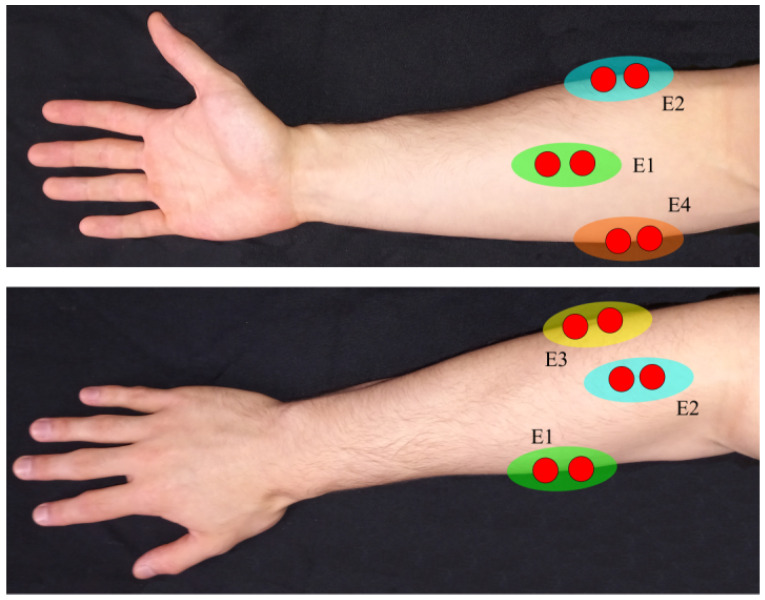
EMG electrode placement on hand muscles. E1 Flexor digitorum superficialis. E2 Extensor digitorum communis. E3 Flexor carpi ulnaris. E4 Flexor pollicis longus.

**Figure 9 biomimetics-08-00029-f009:**
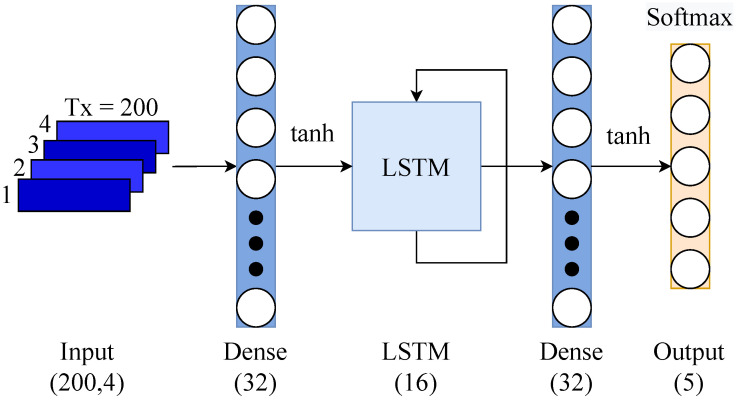
Recurrent Neural Network Architecture.

**Figure 10 biomimetics-08-00029-f010:**
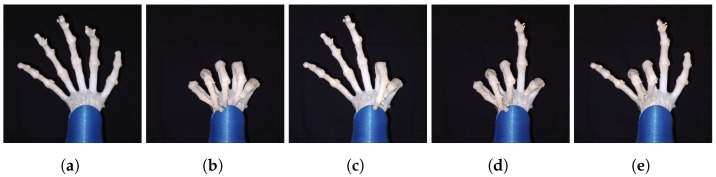
Gestures used by the LSTM-RNN classifier for the EMG control. (**a**) Open hand. (**b**) Closed hand. (**c**) Lateral pinch. (**d**) Signaling sign. (**e**) Rock sign.

**Figure 11 biomimetics-08-00029-f011:**
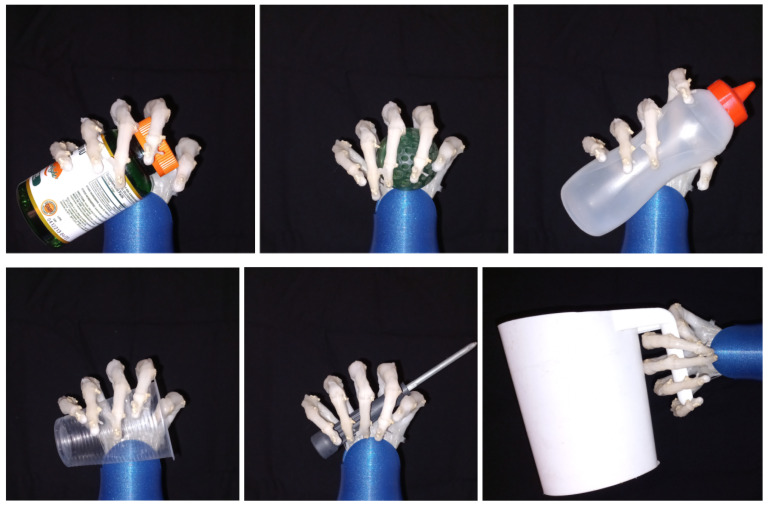
Power grip of different objects.

**Figure 12 biomimetics-08-00029-f012:**
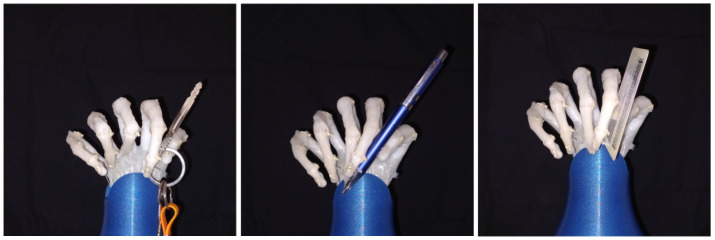
Precision grip of different objects.

**Figure 13 biomimetics-08-00029-f013:**
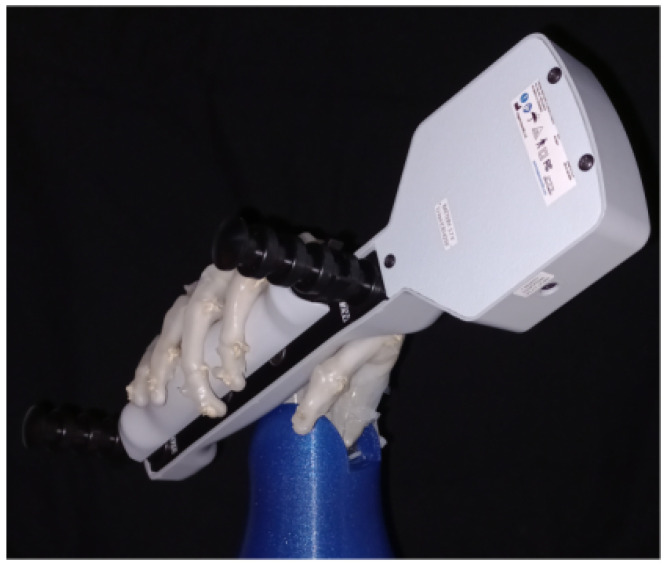
Experimental set up for force measurement.

## Data Availability

The data that support the findings of this study are available from the corresponding author, J.C.Tejada (juan.tejada@eia.edu.co), upon reasonable request.

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
