# Peer review of "A Proposal of Bioinspired Soft Active Hand Prosthesis"

_biomimetics, 2023, doi:10.3390/biomimetics8010029_

Round 1
Reviewer 1 Report
This paper proposes a bionic 14 degree of freedom flexible manipulator, which mainly mimics the skeletal structure and musculoskeletal structure of a human hand, and this bionic design is proven to be adaptable and compliant, providing some design reference for future structural design of humanoid dexterous hands.
From the research framework of this paper, the paper firstly introduces the design of the prosthetic hand, secondly explains the selection of materials and manufacturing process, as well as the control mode of the prosthetic hand, and proves the adaptability and compliance of the prosthetic hand by setting up several groups of test experiments to grasp objects of different shapes and sizes.
The work proposed in this paper show some innovation to this area. However, the paper still has the following shortcomings:
(1) The prosthetic hand uses nylon rope as the tendon, and the performance of the prosthetic hand will certainly deteriorate as the number of trials increases.
(2) The finger cannot move sideways and the range of grasped objects is greatly limited.
(3) EMG control systems are capable of recognizing a limited range of human gestures and have a large time delay.
(4) The experimental test section mainly considered only the application of the grasping aspect and did not carry out experiments about the manipulation aspect.
(5) The pictures related to this paper such as Figure 7 and Figure 10 look confusing, and the caption part is not centered.
Author Response
Juan C. Tejada
Faculty of Engineering
Computational Intelligence and Automation Group (GIICA)
EIA University
Km2+200 ms. Variante Aeropuerto José María Córdova
Envigado, Antioquia
Email: juan.tejada@eia.edu.co
December 14, 2022
Re: Submission of revised version of manuscript
Authors response to Reviewers Comments
Title: “A proposal of Bioinspired Soft Active Hand Prosthesis”
We would like to thank the reviewer for the comments and suggestions to improve the paper.
Comments:
This paper proposes a bionic 14 degree of freedom flexible manipulator, which mainly mimics the skeletal structure and musculoskeletal structure of a human hand, and this bionic design is proven to be adaptable and compliant, providing some design reference for future structural design of humanoid dexterous hands.
From the research framework of this paper, the paper firstly introduces the design of the prosthetic hand, secondly explains the selection of materials and manufacturing process, as well as the control mode of the prosthetic hand, and proves the adaptability and compliance of the prosthetic hand by setting up several groups of test experiments to grasp objects of different shapes and sizes.
The work proposed in this paper show some innovation to this area. However, the paper still has the following shortcomings:
- The prosthetic hand uses nylon rope as the tendon, and the performance of the prosthetic hand will certainly deteriorate as the number of trials increases.
- The finger cannot move sideways and the range of grasped objects is greatly limited.
- EMG control systems are capable of recognizing a limited range of human gestures and have a large time delay.
- The experimental test section mainly considered only the application of the grasping aspect and did not carry out experiments about the manipulation aspect.
- The pictures related to this paper such as Figure 7 and Figure 10 look confusing, and the caption part is not centered.
Answer:
The changes made in the manuscript were highlighted in order to ease the revision of the paper.
Regarding the mentioned shortcomings of the system, we are well aware of the system limitations, however, the aim of the research was more oriented to provide a bioinspired design that could take advantage of the musculoskeletal structure and biomechanical properties of the human hand and present a proof of concept to validate the design. The mentioned short coming are proposed as future work of this research, in particular the addition of thumb abduction which allows a greater dexterity of the system, the addition of more EMG channels in order to recognize more gesture and the use of signal segmentation techniques like sliding window instead of the adjacent window technique; however, it is important to note that increasing the complexity of the EMG control system could significantly increase the computational cost, limiting its potential implementation in an embedded system, which is not desirable for a portable and self-contained device like a hand prosthesis.
The caption of figure 7 and figure 10 were corrected and modified so they could give the reader a better understanding of the image. The MDPI template align the captions to the left as it can be seen in all figures of the manuscript.
The reviewer is again thanked for the comments, and any questions or concerns that may arise regarding the review, the authors will be quick to respond. We hope that our revised paper meets with your satisfaction.
Sincerely yours,
Alejandro Toro-Ossaba, Juan C. Tejada, Santiago Rúa and Alexandro López-Gonzáles.

Reviewer 2 Report
The authors propose a dexterous bioinspired soft active hand prosthesis based on the skeletal architecture of the human hand. The design includes the imitation of the musculoskeletal components and morphology of the human hand, allowing the prosthesis to emulate the biomechanical properties of the hand, which results in better grips and a natural design. They are interesting and impressive to the readers.
1. In the introduction, they should introduce some actuating or sensing methods for the fingers or hands. like pneumatic methods or hydraulic methods. Eccentric actuator driven by stacked electrohydrodynamic pumps or Soft fiber-reinforced bending finger with three chambers actuated by ECF (electro-conjugate fluid) pumps.
2. References 2, and 4 are not correct.
3. A question about the Recurrent Neural Network (RNN) Classifier, why do they use the RNN method?
4. an optical picture like figure.5 is necessary for their paper.
Author Response
Juan C. Tejada
Faculty of Engineering
Computational Intelligence and Automation Group (GIICA)
EIA University
Km2+200 ms. Variante Aeropuerto José María Córdova
Envigado, Antioquia
Email: juan.tejada@eia.edu.co
December 14, 2022
Re: Submission of revised version of manuscript
Authors response to Reviewers Comments
Title: “A proposal of Bioinspired Soft Active Hand Prosthesis”
We would like to thank the reviewer for the comments and suggestions to improve the paper.
Comments:
The authors propose a dexterous bioinspired soft active hand prosthesis based on the skeletal architecture of the human hand. The design includes the imitation of the musculoskeletal components and morphology of the human hand, allowing the prosthesis to emulate the biomechanical properties of the hand, which results in better grips and a natural design. They are interesting and impressive to the readers.
- In the introduction, they should introduce some actuating or sensing methods for the fingers or hands. like pneumatic methods or hydraulic methods. Eccentric actuator driven by stacked electrohydrodynamic pumps or Soft fiber-reinforced bending finger with three chambers actuated by ECF (electro-conjugate fluid) pumps.
- References 2, and 4 are not correct.
- A question about the Recurrent Neural Network (RNN) Classifier, why do they use the RNN method?
- an optical picture like figure.5 is necessary for their paper.
Answer:
The changes made in the manuscript were highlighted in order to ease the revision of the paper.
An new paragraph was added in the introduction, introducing common actuation methods used in soft robotic fingers, including pneumatic (PneuNets) and hydraulic; the two research papers suggested by the reviewer were included in this paragraph.
References 2 and 4 were corrected.
Taking into account that the EMG signal is a time series signal, we selected the RNN architecture because this type of deep learning model has proven to perform well recognizing patterns in time series data, particularly, the LSMT units have the ability to retain long and short term dependencies of the time series allowing the model to relate pass information of the signal with current one, allowing for a better prediction. The reviewer is invited to read our research about the LSTM-RNN gesture classifier used in this paper. LSTM-RNN research: https://doi.org/10.3390/app12199700.
The proposal picture has been included like graphical abstract, thank for the recommendation.
The reviewer is again thanked for the comments, and any questions or concerns that may arise regarding the review, the authors will be quick to respond. We hope that our revised paper meets with your satisfaction.
Sincerely yours,
Alejandro Toro-Ossaba, Juan C. Tejada, Santiago Rúa and Alexandro López-Gonzáles.

Round 2
Reviewer 1 Report
The authors have addressed all the questions I concerned. Although I think some of the shortcomings I mentioned last round can still be improved, I agree with the explanation by the authors in their future work.
Reviewer 2 Report
I thank the authors for making such an excellent effort to incorporate the recommendations from both myself and the other reviewer. The quality of your responses to our comments is excellent. I feel the paper has gained a lot in terms of quality and clarity.